# A Potential Approach of Reporting Risk to Baseflow from Increased Groundwater Extraction in the Murray-Darling Basin, South-Eastern Australia

**Glen Walker**

Grounded in Water, Adelaide, SA 5064, Australia; glen.walker@internode.on.net

**Abstract:** An approach of reporting long-term trends in groundwater extraction and baseflow impacts in the Murray-Darling Basin (MDB) in south-eastern Australia was developed and tested. The principal aim of the framework was to provide early warning of any potential adverse impacts from groundwater extraction on environmental releases of surface water for baseflow, support adaptive management of these impacts, and highlight those areas which may benefit from conjunctive water management. The analysis showed that there is no current decadal trend in the annual aggregate groundwater extraction volumes or stream impact across the non-Victorian MDB, with much of the interannual variability being related to rainfall. Despite this, increasing volumes of environmental releases of water for baseflows in some river valleys are being required to replace the stream depletion caused by historical patterns of groundwater extraction established before 2003. Two valleys were identified for which there may be insufficient surface water storage to release water to substitute stream losses to groundwater and still support ecosystems during dry periods. The increasing trend in extraction since 2003 in one of the units has significantly increased the risk in that valley. The reporting framework was shown to be effective for alluvial groundwater systems connected to regulated rivers.

**Keywords:** Murray-Darling Basin; groundwater extraction; stream depletion; environmental baseflow; connectivity; reporting; trend analysis; risk indicators

## 1. Introduction

The connected nature of surface and groundwater resources can lead to difficulties in meeting ecological and water quality and quantity objectives for rivers in drier regions, where groundwater extraction forms a significant fraction of the diversion of water [1–11]. This can be exacerbated by a drying climate or more prolonged droughts caused by climate change, as this can mean extended periods of reduced surface water availability and increased pressure on groundwater [6,7].

The Murray-Darling Basin in south-eastern Australia has low run-off and large variability of flow [12,13]. Its water management plan (the plan) aims to rebalance consumptive use of water with environmental needs by reducing limits on diversions (Sustainable Diversion Limits—SDLs) of surface water for each river valley [14]. While SDLs for groundwater have also been implemented for eighty groundwater units across the whole of the MDB, including for aquifers at various depths, the aggregate SDL for groundwater is two to three times greater than current groundwater extraction [15]. Concerns have been raised [16,17] that an increase in the volume of groundwater extraction would lead to a potentially large impact on surface water availability, thus undermining the efforts to recover water for the environment.

There have been several studies that have projected the cumulative impact of increased groundwater extraction on streamflow in the MDB [15,18–24]. The most recent

of these [15] examined the impacts of extractions under the Plan and found that most groundwater extraction into the next two to three decades is likely to occur through use of entitlements and rights that existed before the Plan. Additionally, there is high uncertainty in the timing and location of any further increases in extraction from the remaining 'unassigned water', despite the pressures of a drying climate [25]. The Murray-Darling Basin Authority (MDBA), which together with Basin states administers the Basin Plan, notes that the increase so far has occurred in groundwater systems with lower connectivity to streams; but if extraction increases in groundwater systems with moderate to high connectivity with streams, the impacts on surface water resources might be significant [26]. Because of the long time for such increases to be realized and the large uncertainties, the MDBA 'will continue to monitor and improve our knowledge base accordingly to support the identification and management of risks'.

This paper explores an approach to reporting that would support an active adaptive risk management framework on this issue. The challenge in using the adaptive management approach lies in finding the correct balance between gaining knowledge to improve management in the future and achieving the best short-term outcome based on current knowledge. The study assumed that any such approach would rely on the following characteristics, as a minimum:

1.  The desired surface water flow outcomes are expressed in the management objectives of the Plan and transparently monitored across the MDB;
2.  There is a transparent reporting of the actions, i.e., groundwater extraction across the MDB;
3.  There is a quantitative or semi-quantitative link between the actions and the sought outcomes;
4.  The information should be able to guide further studies and policies.

Maintaining hydrological regimes is an important objective of the Basin Plan and Basin-wide environmental watering strategy [27,28]. This includes baseflows, which are sensitive to groundwater extraction. In this context, baseflows are reliable background flow levels within a river channel that are generally maintained by seepage from groundwater storage and also by surface water inflows [28]. One agreed measure of success of the Plan is that baseflows are maintained at least 60% of the natural level [27]. However, low-flow metrics, such as the 60% criterion, are recognized for not adequately representing ecological needs throughout the river system. Additionally, the 'natural' baseline is difficult to assess without an appropriate 'natural' model that represents low-flow processes [27]. There has been a shift to defining baseflow thresholds (volumes, timing, recurrence intervals) at gauging stations along regulated reaches of major river valleys [29], which are not only relevant for ecological function, but are more easily measured. These thresholds guide surface water management, especially release of environmental water from storage [29]. While there has been an expectation of delivering baseflows from surface water storage [29], a recent analysis [30] found that there were no northern Basin catchments where the Basin target was met annually during the 2014–19 period.

River management models could, in principle, be used to estimate risk to meeting baseflow thresholds from increased groundwater extraction. However, the state of the current models may not adequately represent low flows [31,32] or changing groundwater inflows to streams [33]. In the absence of these, other approaches are required. Trends in groundwater extraction, as indicated by annual reporting of groundwater extraction [34,35], can be linked to the change in mean annual streamflow through a connectivity factor, CF [15,36], where the value of CF lies between 0 (disconnected) and 1 (full connectivity). CF can be determined through numerical and analytical models or based on conceptualization [36]. The determination of CF across the MDB to provide reliable extrapolations of risk is inhibited by the sheer size and hydrogeological complexity of the

MDB. Furthermore, the large time lag, possibly decades, between changes in extraction and streamflow significantly affects the efficacy of adaptive management [37].

This paper aimed to develop and test the applicability of a reporting approach to support adaptive management of this issue. It requires as inputs the annual groundwater extraction volumes and monitoring of groundwater levels and streamflows. It provides outputs relevant to 1) early warnings of increasing groundwater extractions and impacts on streamflow; 2) volumes of stream loss caused by increasing groundwater extraction; and 3) identifying reaches, where conjunctive water management may be required to support baseflows by protecting connectivity between groundwater and surface water. The approach consists of five building blocks. The output of the first four steps is an estimate of impact, with uncertainty, from historical extraction patterns and relies on annual reporting of extraction volumes and groundwater modelling outputs.

The five elements are:

1. *A first-step prioritization of units*: previous studies have indicated that the nature of the MDB meant that only a limited number of groundwater units were undergoing sufficient increases in extraction to lead to significant impacts. While there was a 53% increase in the annual extraction volume from 2012–2013 to 2018–2019, much of this increase occurred in a minor proportion of groundwater units, in which there are groundwater models and reasonable groundwater information. It is proposed that there should be a focus on these units as a way of balancing the need for further studies with the need for action. As extraction in other units begin to increase, further information can be developed as required. The choice of the number of units should be such to allow early identification of units where extraction may be emerging as a problem.

2. *Decadal trend analysis:* for the priority units identified in Step 1, a decadal trend analysis of extraction volumes was conducted. Much of the increased extraction over the period of 2012–2013 to 2018–2019 is likely to be related to short-term processes, such as rainfall variability [38]. A trend analysis separates these processes from long-term increases in groundwater extraction. Such long-term increases can lead to a range of risks, but in the context of this paper, to baseflow. The short-term variations in extraction are important in themselves for the management of water resources.

3. *Determination of CF and associated attributes of the groundwater system*: the estimation of CF and other attributes, such as time lags, depends on the hydrogeological characteristics and data availability of the priority units.

4. *Determination of the impacts on mean flows:* the aim of Step 4 is to determine impacts on streams using outputs from the previous two steps. A risk analysis needs to incorporate the large uncertainties associated with both the analysis of trend and the estimation of CF. The large time delays associated with groundwater systems mean that the increases in extraction need to be placed in a historical context with changes possibly still occurring in groundwater systems from extraction patterns established twenty years ago.

5. *Indicators of the significance of the impacts*: the estimated impacts in step 4 need to be placed in context of environmental baseflow objectives. Identification and testing of indicators of risks to baseflow should be conducted for various regulated river valleys in the MDB. It is expected that the risk is higher in the northern MDB, where groundwater extraction is proportionately larger and flows tend to be less regulated and more ephemeral than in the southern MDB. Problems with protecting the longitudinal connectivity of baseflows has led to major issues of fish kills in the northern MDB [36].

While ideally, the approach should be tested for the whole of the MDB, this was difficult for various reasons. Steps 1 and 3 were conducted for the whole of the MDB, but Step 2 could not be fully applied to Victoria because of the length of record of readily

available data. This is being addressed outside of this paper. Step 4 was only applied to all but the southern connected system, which includes Victoria. Step 5 was applied to the river valleys in New South Wales, as these have clearly defined baseflow targets on regulated reaches. Because of inherent time lags and the low number of units with increasing extraction, the testing involved all historical groundwater extraction and not just the recent trends. Despite not testing the approach for the whole of the MDB, the results from those river valleys that went through all five steps provide a test of the reporting framework.

## 2. Material and Methods

### 2.1. Study Area

This section briefly describes the surface and groundwater hydrology of the MDB; history of groundwater management changes; the implementation of the Plan; and environmental water strategies. More comprehensive descriptions of the MDB are available [39–42].

The MDB, shown in Figure 1, drains a little over 1 million km² of inland Australia in the south-east of the continent, including parts of the states of Queensland (Qld), New South Wales (NSW), Victoria (Vic.), and South Australia (SA) as well as the Australian Capital Territory. The upstream extent of the MDB is bounded by the Great Dividing Range, which runs roughly parallel to Australia's coastline in the south-east of the continent. The MDB's larger tributaries drain upland areas along the inland side of the Great Dividing Range, heading generally towards the dry centre of the Basin. These tributaries quickly pass from upland valleys and traverse the Darling or Murray riverine plains before joining one of the Basin's two major rivers. The northern tributaries, including the Condamine (Qld), Gwydir, Namoi, Macquarie, Castlereagh, and Bogan (NSW), join the Darling River while the southern tributaries, including the Murrumbidgee (NSW), Murray (NSW and Vic.), Goulburn, and Broken (Vic.), join the Murray River; with the Darling River joining the Murray River close to the border of Victoria, NSW, and South Australia. The Murray River discharges into the Southern Ocean in South Australia after passing to the east of the Mt Lofty Ranges. The Lachlan River (NSW) only joins the southern basin during flooding, while most of the streams off the Mt Lofty Ranges only join the Murray River during wetter periods.

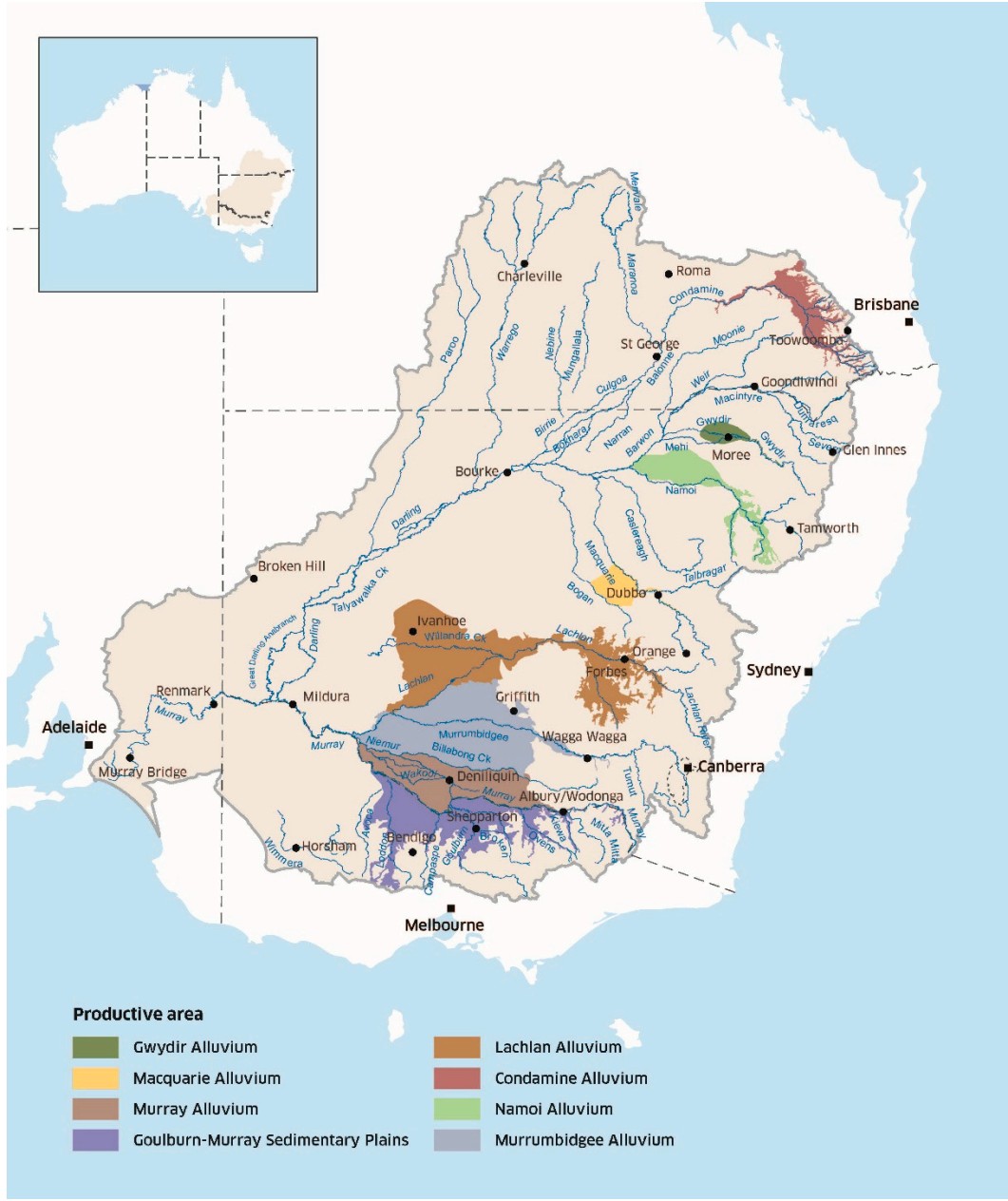

**Figure 1.** Map of the Murray-Darling Basin showing the alluvial groundwater systems (Source: MDBA). The other priority units include EMLR (east of Adelaide), MGL (south of Renmark), GMH (north of Melbourne and UCB (south-east of Toowoomba). The states containing Adelaide, Melbourne, Sydney and Brisbane are respectively South Australia, Victoria, New South Wales and Queensland.

The MDB has generated an aerial average of just 28 mm/year of streamflow over the period of post-European record, which is only 6% of rainfall [43]. The hydrology of the MDB is amongst the most variable in the world, even compared to other arid zone basins [12,13]. In particular, the MDB has experienced long multi-year droughts recently including the Millennium Drought (1997–2010) and another period of severe drought over the last three years (2017–2019). Climate modelling projects increases in the frequency and severity of droughts and an overall decline in streamflows with global warming; a trend which is consistent with trends in recent decades [44–47].

The main sources of surface flow are the south-eastern and eastern boundaries of the southern MDB [21], producing close to half of average inflows and providing a more

secure water supply for towns and permanent irrigated crops. Tributaries in the northern MDB have lower yields and higher variability from year to year. The southern MDB was developed earlier and includes the MDB's largest dams and significantly greater areas of irrigated agriculture. The rivers experience high transmission losses with approximately 41% of streamflow entering the MDB rivers flowing to the sea prior to development [21].

The Murray-Darling Basin overlies a variety of geological units [48]. The northern MDB is directly underlain by alluvial sediments up to 200 m thick that in turn overlie the Mesozoic deposits of the Great Artesian Basin (GAB) which comprise sandstones and mudstones of the Jurassic to Cretaceous age. Although the GAB is a vital resource for much of inland Australia, it is covered by a separate management arrangement and is not considered further here. The south-western portion of the Murray-Darling Basin is underlain by the Murray Geological Basin, a thin assemblage of flat-lying horizontally bedded fluvial and shallow marine sediments of the Tertiary age. These sediments vary in thickness from less than 200 m in the north, east, and south to 600 m in the west central part of the Basin [48,49].

The groundwater systems occur in a range of hydrogeological settings but can be subdivided into three major provinces:

1.  Fractured rock aquifers: these of the Mt Lofty and Flinders Ranges and the Great Dividing Range contain fractured rock aquifers of moderate productivity.
2.  Major alluvial systems: these have been formed from the deposits of sand and gravel from the main river tributaries and are the source of most groundwater extraction in the MDB. The major units are shown in Figure 1.
3.  Tertiary limestone of the western Murray Geological Basin: good-quality groundwater in this aquifer was recharged tens of thousands of years ago during a wetter climate.

The streams in the narrow alluvial reaches are closely connected with adjacent shallow aquifers that are constrained by bedrock and are mostly gaining under natural conditions. Further downstream, the river valleys open into wider riverine plains and groundwater levels drop away, resulting in a shift to losing conditions. Along the Murray and Darling Rivers, bedrock highs and low aquifer transmissivity force groundwater levels near the surface again, and hence the major rivers tend to be neutral or gaining. River leakage and flooding are major sources of recharge and have led to areas of fresher groundwater in the otherwise brackish to saline regional groundwater. This overall predictable pattern is followed with minor deviations along the major tributaries [50,51].

Groundwater development was encouraged after WW2 to mainly support irrigation. During the 1970–1990 period, it was clear that some groundwater systems were becoming stressed. In 1996, the strategic framework for Australian water reforms was amended to include groundwater. In 2004, the Council of Australian Governments (COAG), under the National Water Initiative (NWI) [52], committed to prepare comprehensive water plans, achieve sustainable water use in over-allocated or stressed water systems (including groundwater), and expand trade in water rights. The right to the use, control, and flow of water is vested in state governments, and water rights can be withdrawn or altered. Planning is undertaken by the water resources agency to allocate water between consumptive and non-consumptive uses based on an assessment of economic, social, and environmental benefits and costs. The plans are adaptively managed to provide flexibility to address regulatory error, new scientific evidence, and changing community values and allow large changes to allocations to be implemented gradually. The trading of water rights (permanent trades) or of water flows (temporary trades) has been encouraged to facilitate re-allocation of water from lower- to higher-valued uses, increasing the benefits obtained from the scarce resource.

During the 2000s, there were efforts to address stressed systems. For example, entitlements for seven alluvial groundwater systems in NSW (Lower Murray, Lower

Murrumbidgee, Lower Lachlan, Lower Macquarie, Lower Namoi, Upper Namoi, and Lower Gwydir) were reduced under the Achieving Sustainable Groundwater Entitlements (ASGE) program, and groundwater sharing plans were developed for these [21]. In Victoria, water level response management was implemented in some major groundwater systems, such as Katunga in the Goulburn-Murray Sedimentary Plain. This meant that the annual water determination (and hence groundwater use) was often less than the full entitlements [53].

Under the NWI, the Australian governments made an undertaking to manage connected systems as a single resource, including 1) the identification of sites of close interaction between groundwater aquifers and streamflow; 2) the development of common arrangements; and 3) the development and implementation of systems to integrate the accounting of groundwater and surface water use. These expectations were included in the legislation supporting the Basin Plan. Increased groundwater extraction was identified as one of six risks [54] to streamflows (climate, farm dams, afforestation, irrigation efficiency improvements, groundwater extraction, and bushfires) in which drivers outside of the normal regulatory system could affect water availability and hence undermine efforts to limit surface water diversion. This led to cumulative impact assessments for groundwater mentioned earlier that eventually led to the assignment of a low priority for risk from increased groundwater extraction in the development of the Plan.

In 2007, the Australian government allocated AUD 10B to recover water for the environment from the consumptive water pool. Since 2008, the Commonwealth has been acquiring environmental water in two ways: first by buying permanent water entitlements (rights) from irrigators, and second by investing in on-farm and off-farm infrastructure modernization projects, with part of the water 'saved' reverting to the Commonwealth. This environmental water is called 'held' water, and the entitlements have the same conditions on them as for consumptive entitlements. The only recovery of groundwater entitlements was in the Upper Condamine Alluvium in Queensland [24].

The MDB Plan was introduced in 2012 and implemented in 2019–2020. It was the first time that limits on groundwater use were put in place across the entire Basin (in contrast to surface water, where the cap on surface water diversions has been in force since the late 1990s. It was also the first time a consistent set of management arrangements were applied across all the Basin's groundwater resources, with the exception of the Great Artesian Basin [24]. Groundwater units were defined, mainly based on hydrogeology, groundwater flow, and management arrangements within each jurisdiction. The large aggregate extraction limit largely reflects parts of the MDB, where a groundwater extraction limit that was not previously defined now existed. A Baseline Diversion Limit (BDL) was also defined for each unit to provide the baseline against which SDLs are assessed. This limit reflects the plan limit or level of entitlement for where a plan existed before the MDB Plan; and the entitlement volume along with the effect of any rules managing extraction for where there previously was no plan. Where the SDL exceeds the BDL, the difference is referred to as unassigned water.

Conjunctive water management has been implemented in the water plans for 'highly connected' systems, where the response times from extraction on streams are less than one year and the alluvial systems are narrow and shallow [55,56]. The management of systems with slower responses is largely through the determination of the SDL [55] or water table response management [53], supported by measures such as distance rules and specific management zones [26].

Groundwater trade has developed more slowly than surface water trade with about 180 GL/y of entitlement trade in the MDB in 2018–2019 and 89% of all trade in 2018–2019 in the MDB occurring in New South Wales and about 40% in the Lower Murrumbidgee groundwater unit. Within the Murrumbidgee catchment, trade was used to replace

surface water with groundwater as an irrigation source when surface water availability was reduced [57]. There is little managed aquifer recharge in the MDB [30].

The MDB Plan required the establishment of a Basin-wide environmental water strategy [27,58,59]. By 2019, the water recovery had led to around 2851 GL of environmental water entitlements 'held' by the by the Commonwealth Environmental Water Holder (CEWH) and was expected to yield on average 1978 GL/year of water across the MDB [60]. Including additional volumes held by state governments, the total environmental water holdings in the MDB is approximately 30% of the SDL for the MDB [59]. In addition to the water held in entitlements (both consumptive and 'held' environmental), there is additional 'unallocated' water known as 'planned' environmental water, the largest volume of environmental water by far. The use of this 'planned' water is directly controlled under the terms of the water resource plans, and its availability is dependent on the rules set out in each water resource plan. As such, it becomes more vulnerable to climate change than held environmental water [61,62].

*2.2. Methods*

*1. First-step prioritization*: in the first step, fifteen units were chosen for further work based on being the first fifteen largest differences between the years 2012–2013 and 2018–2019, using data from the MDBA transitional reports [35]. Other units were then reviewed as to whether temporal variations were larger than 5 GL/yr, even if the difference for the first and last year of the sequence was not large. On this basis, another three units were added. These units are collectively described as 'priority units' within this paper and are shown in Table 1. The units represent almost 95% of the increased extraction; 82% of the average extraction from 2012–2013 to 2018–2019; and 49% of the extraction limit. The average extraction from 2012–2013 to 2018–2019 represents 68% of the extraction limit for these units. In 2018–19, 92% of the total annual actual take from groundwater was reported to be metered, while 100% of take under basic rights (~12%) was unmetered [35].

Fourteen of the eighteen priority units are alluvia, while three are fractured rock systems and one a Tertiary limestone unit. This list included all groundwater systems, for which either entitlements had been reduced or the extraction limit was limited by an annual water determination, including all ASGE units, the Goulburn Murray Sedimentary Plain, and the Upper Condamine Alluvium.

Apart from these 'usual suspects', there are nine other groundwater units. Three are broad-valley-constrained floodplain alluvia (Upper Lachlan, Upper Macquarie, and mid-Murrumbidgee). Such alluvia usually have a higher CF and generally have developed later than the riverine plain alluvia. In addition, there are two shallow alluvia of the southern Riverine Plain (Shepparton irrigation area and Lower Murray Shallow Alluvium). These shallow units have been traditionally pumped to minimize land salinity and waterlogging. However, drier climates over the last twenty or so years have meant that these have been mainly used as a resource. There are two fractured rock systems, the Upper Condamine Basalts in Queensland and the Goulburn-Murray Highlands in Victoria. The final two units are the eastern Mt Lofty Ranges (a mix of fractured rock and Tertiary limestone) and Murray Group Limestone in South Australia. Details of these groundwater units can be found in [63–72].

**Table 1.** The fifteen groundwater SDL units with the largest difference between extraction volume during the 2012–2013 and 2018–2019 years and an additional three units with large variations during this time. Additionally shown is the state in which the unit is found, their groundwater type, the extraction volumes in 2012–2013 and 2018–2019 in GL, the difference between these, the BDL and SDL in GL/year, and the average annual extraction volume over the period. The alluvial groundwater types are denoted by RP (Riverine Plain), RP-D (deeper units of the Riverine Plain), RP-S (shallow unit of Riverine Plain, BVF (broad-valley-constrained floodplain), and NVF (narrow-valley-constrained floodplain). FR denotes a fractured rock unit.

| Ranking/ID | Unit | State | Groundwater Type | Annual Groundwater Extraction 2012–2013 (GL/year) | Annual Groundwater Extraction 2018–2019 (GL/year) | Difference between 2012-2013 and 2018–2019 (GL/year) | BDL (GL/year) | SDL (GL/year) | Average between 2012–2013 and 2018–2019 (GL/year) |
|---|---|---|---|---|---|---|---|---|---|
| 1/LMbD | Lower Murrumbidgee–Deep Alluvium | NSW | RP-D | 179.6 | 377.9 | 198.3 | 273.6 | 273.6 | 261.6 |
| 2/LN | Lower Namoi Alluvium | NSW | RP | 61.1 | 116.2 | 55.1 | 88.3 | 88.3 | 89.5 |
| 3/SIR | Goulburn Murray–Shepparton Irrigation Region | Vic | RP-S | 41.3 | 96.3 | 55 | 244.1 | 244.1 | 56.3 |
| 4/LMD | Lower Murray–Deep Alluvium | NSW | RP-D | 56.2 | 110.7 | 54.5 | 88.9 | 88.9 | 68.7 |
| 5/GMSP | Goulburn-Murray–Sedimentary Plain | Vic | RP-D | 101.2 | 149.1 | 47.9 | 203.5 | 223 | 126.6 |
| 6/UL | Upper Lachlan Alluvium | NSW | BVF | 44.2 | 89.4 | 45.2 | 94.2 | 94.2 | 57.4 |
| 7/LL | Lower Lachlan Alluvium | NSW | RP | 87.2 | 131.8 | 44.6 | 123.4 | 117 | 108.6 |
| 8/CC | Upper Condamine Alluvium (Central Condamine Alluvium) | Qld | BVF | 32.3 | 57.7 | 25.4 | 81.4 | 46 | 46.7 |
| 9/UN | Upper Namoi Alluvium | NSW | BVF | 90.1 | 112.2 | 22.1 | 123.4 | 123.4 | 98.3 |
| 10/UMq | Lower Macquarie Alluvium | NSW | RP | 26.9 | 47.4 | 20.5 | 52.7 | 52.7 | 32.9 |
| 11/MMb | Mid-Murrumbidgee Alluvium | NSW | BVF | 35.5 | 55.6 | 20.5 | 53.5 | 53.5 | 39.0 |
| 12/LMS | Lower Murray: Shallow Alluvium | NSW | RP-S | 2.26 | 11.9 | 9.6 | 14.1 | 14.1 | 6.3 |
| 13/Umq | Upper Macquarie Alluvium | NSW | BVF | 13.7 | 23 | 9.3 | 17.9 | 17.9 | 16.6 |
| 14/EMLR | Eastern Mt Lofty Ranges | SA | TL/FR | 2.83 | 11.6 | 8.8 | 34.7 | 38.5 | 5.7 |

| 15/LG | Lower Gwydir Alluvium | NSW | RP | 29.3 | 37.5 | 8.2 | 33 | 33 | 35.9 |
| 16/GMH | Goulburm-Murray Highlands | Vic | FR | 9.9 | 15.5 | 5.6 | 38.3 | 68.7 | 14.3 |
| 17/UCB | Upper Condamine Basalts | Qld | FR | 65.1 | 58 | −7.1 | 79 | 79 | 65.3 |
| 18/MGL | Murray Group Limestone/MGL | SA | TL | 41 | 38.7 | −2.3 | 63.6 | 63.6 | 36.0 |
| Sub-total (18) | | | | 919.7 | 1540.5 | 621.2 | 1707.6 | 1719.5 | 1165.7 |
| Total | | | | 1223.2 | 1882.4 | 659.2 | 2365 | 3472 | 1415.3 |
| Non-priority units | | | | | 38.0 | 657.4 | 1752.9 | 249.6 |

*2. Determining trends*: the above extraction data for the priority units were augmented by publicly available data for as much of the period from 2003–2004 to 2019–2020 as possible. Data for all units are available for 2012–2013 to 2019–2020, and most units outside of Victoria have data for a longer period. As part of the Basin Plan development, boundaries of the Victorian units were changed, meaning that data before 2012–2013 are not readily available. For two NSW units (LMb and LM), aquifers were separated in 2008–2009, with only combined data available before then. For these, the extraction for the shallow aquifer before 2008–2009 was estimated for this paper. Earlier extraction data tend to be less reliable than recent data, as estimation methods have been made increasingly more consistent and a higher level of metering has been introduced.

The extraction data for the priority units were analysed using trend analysis. For these analyses, extraction data were normalized with respect to the SDL. For some units, a supplementary license was used to transition with the prior extraction limit to the final SDL but was not included in the definition of the SDL here. A trend and confidence and prediction intervals were determined using the least square algorithms (LINEST and regression) and the 'tinv' function in Excel for (a) each priority unit during the 2012–2013 to 2018–2019 period; (b) each priority unit for the maximum time period of available data; (c) the aggregate extraction from the priority units for 2012–2013 to 2019–2020; and (d) 80 MDB units for 2012–2013 to 2019–2020. The slopes and standard errors are reported.

A meteorological station on the Bureau of Meteorology database was identified for each groundwater SDL unit and rainfall collated for the 2003–2004 to 2019–2020 period. The rainfall was normalized according to

$$P' = (P - P_{mean})/P_{mean}, \tag{1}$$

where $P_{mean}$ is the annual mean for the available meteorological station for the time period of the available data; P is the annual rainfall; and P' is the normalized rainfall. The mean was estimated for the whole of the MDB by simply taking the mean of the normalized data from each rainfall station. The time series for each unit and the mean for the MDB are plotted to visualize the rainfall variability across the MDB. As the priority groundwater units are found in semi-arid to sub-humid climate zones, the mean rainfall reflects these zones rather than the whole MDB.

Relationships of the annual extraction volume with rainfall and time were explored to interpret increases in groundwater extraction. Two different approaches were used to separate long-term increases in extraction from rainfall variations. The first was a

graphical approach, shown in Figure 2. The normalized annual extraction is plotted against scaled annual rainfall and labelled by the number of years since 2003–2004. By following vertical lines representing different rainfall values, one can identify whether long-term extraction is increasing. For example, by following the *y*-axis (mean rainfall), one can see whether the mean annual extraction is increasing. A stable extraction pattern leads to a cycle from the top left-hand side to the lower right-hand side with no net change in extraction across the *y*-axis. Previous studies have shown that annual extraction [38] can often be linear with annual rainfall. However, there are several different influences on extraction, including commodity prices, government policy and planning, groundwater trading, climate changes, and rainfall in preceding years.

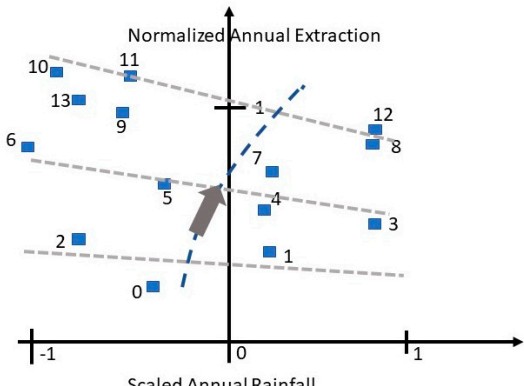

**Figure 2.** Schematic showing how relationships between groundwater extraction and rainfall over time can be conceptualized. The data labels show the number of years since 2003–2004, and the dotted line shows the trajectory of the trend line for groundwater extraction with rainfall over time. If extraction is increasing in time, there are upward shifts. By following a single rainfall value, e.g., x = 0 (the mean rainfall), the effect of rainfall variations is removed. Lower rainfall generally leads to increased extraction and hence there is a shift to the upper left-hand side for drier climates. A stable extraction cycle that is stable is represented by a stable trend line.

The second approach was the implementation of either a multiple or linear least-squares linear regression on the years since 2003–2004 with scaled annual rainfall as the dependent variables. Slope (and confidence interval) and predicted extraction (and confidence interval) are reported. This approach works best where there is a linear relationship between extraction and rainfall and where the time series being used is sufficiently long to capture rainfall variability. The slope of the extraction–time relationship has two components. The first is the long-term dependence sought in this study. The second is the implicit components due to extraction being dependent on other variables, especially rainfall, that vary over time. Where the record is sufficiently long, the implicit component associated with variable rainfall is expected to become less significant.

The derived predictors from both the single and double regressions of long-term trend for the different priority units were then aggregated to provide a long-term reference extraction pattern for the period of 2003–2004 to 2019–2020. Some units may need to be excluded if the time series of available data is insufficiently long to provide confidence in the reference. The extraction in groundwater units was classified as increasing or decreasing if the absolute values were greater than the 95% confidence

interval and conditionally increasing or decreasing if absolute values fell between the 95% and 90% confidence intervals. Otherwise, the extraction was classified as neutral.

*3. Determining CF and associated parameters*: the methodology for determining CF depends on the type of the groundwater system and the availability of data, knowledge, and models. One of the benefits of using a prioritization system is that fourteen priority units are alluvial groundwater systems for which a numerical groundwater model has been implemented to support the determination of the SDL. CF is estimated [36] from a linear regression between the difference in streamflow of two scenarios and the difference in the extraction volume that causes it. In practice, both recharge and extraction volumes are modified between scenarios. A reduction in recharge is treated as the same as an increase in extraction with which this assumption is then tested. The slope (CF), the confidence intervals, and $R^2$ are reported. The fraction of the change in streamflow that is captured to discharge to the stream, rather than induced recharge from the stream, is also reported as this can cause increased solutes to the stream and support GDEs near the stream. The sensitivity to extraction, $S_e$, was estimated by

$$S_e = - (E_b/Q_{sb}) \times CF, \tag{2}$$

where $E_b$ and $Q_{sb}$ are, respectively, the extraction volumes and the net groundwater flux to stream under the baseline scenario, and CF is the estimated connectivity from the analysis.

The models used to derive the output are a combination of models developed by the NSW government and as part of the CSIRO MDBSY project [21,73–78]. The outputs [77–88] were produced as part of the technical support for developing the Basin Plan. While there were some variations in scenarios, they were mostly the same and are detailed in Table 2. Scenario 3 was generally used as the baseline sand compared to all possible scenarios. Uncertainties in using the models for determining CF, apart from normal modelling issues of calibration and conceptualization, include 1) models being not specifically developed for surface–groundwater interactions, 2) age of the models, and 3) poor representation of the confined system [36]. For this reason, the range of CF was broadened for the risk analysis to include model predictive uncertainty.

**Table 2.** Descriptions of the scenarios used in the modelling. Refer to the modelling reports for more details on the extraction limits used and the climate recharge.

| | |
|---|---|
| 1 | No extraction with median fifteen-year period within the historical climate. The scenario was repeated—i.e., re-run using the predicted 2060 groundwater elevations from the first run as the initial conditions for the second run—to allow groundwater levels to recover over a longer period. |
| 2 | Previous plan extraction for the full 50 years of model duration. Climate inputs were based on the median fifteen-year period within the historical climate. |
| 3, 3a | Preliminary extraction limit (PEL) starting from 2017 with median fifteen-year period within the historical climate. Scenario 3 is equivalent to Scenario 2 except that from 2017, the extraction rate was set at the PEL rather than the limit specified in the previous plan. Scenario 3a represents a modification to the PEL if the PEL could not be sustainably applied (e.g., southern Riverine Plain model) |
| 4 | Extraction at PEL starting from 2017 with median fifteen-year period within the historical climate and revised spatial distribution of extraction bores. Scenario 4 is equivalent to Scenario 3 except that the spatial distribution of extraction was revised to limit drawdown at key indicator sites. |
| 5 | Extraction at PEL starting from 2017 with dry fifteen-year period within the dry future climate. Scenario 5 is equivalent to Scenario 3 except that the climate inputs for Scenario 5 are representative of a drier climate. |

| 6 | Extraction at PEL starting from 2017 with dry fifteen-year period within the median future climate. Scenario 6 is equivalent to Scenario 3 except that the climate inputs for Scenario 6 are representative of the median climate change projection—i.e., drier than Scenario 3 yet wetter than Scenario 5. |
|---|---|
| 7 | Extraction at PEL starting from 2017 with 30 percent reduction in irrigation recharge. |
| 8 | Extraction at PEL starting from 2017 with 60 percent reduction in irrigation recharge. |
| 9 | Increased extraction limit starting from 2017 with median fifteen-year period within the historical climate. Scenario 9 is equivalent to Scenario 3 except for increased extraction from 2017. |

CF was assigned a value of zero for the MGL unit, as the major groundwater extraction occurs 1) approximately 100 km from the Murray River; 2) regional discharge areas are present down-gradient of the unit; and 3) there are no nearby tributaries. The two fractured rock systems (GMK and UCB) have been assigned a value of CF between 0.5 and 1.0 to represent a higher range of connectivity. Streams in these areas are generally gaining, with relatively high baseflow indices [50,51]. While there are uncertainties in CF associated with the large areas of these units and the nature of the fractured rocks, available connectivity studies and estimates of the baseflow index indicate the connectivity is likely to be high. For this reason, a range of values (0.5–1.0) was assigned to CF for these units. The EMLR is a mixture of the fractured rock and Tertiary limestone, usually in proximity of the streams. As a precautionary approach, CF was assigned a range of 0.5 to 1.0.

*4. Assessing impacts on streamflow*: the developed approach should provide an early warning system for increased extraction and assess the risk from increased extraction on environmental baseflows. The reduction in streamflow, R, from increases in extraction on streamflow, $\Delta E$, could be calculated from:

$$R = -\Delta Q_s = -CF \times \Delta E. \tag{3}$$

As the aim for this paper was to provide information for the risk analysis, it was important to capture information on the uncertainty in CF and in the change in extraction rate. Two different components of the impact were considered, that from the historical increase in extraction, $R_{sh}$, and that from the change in extraction, $R_{sch}$, over the period of 2003–2004 to 2019–2020 or the equivalent for each unit. For the former, the range of extractions was estimated from the trend analysis by the maximum and minimum values of $E_i$ and $E_f$, respectively, considering the standard error and the range of CF. More explicitly,

$$R_{sh}^{be} = \max(E_i, E_f) \times CF^{be}$$

$$R_{sh}^{max} = (\max(E_i, E_f) + CI_{end}) \times \max(CF) \tag{4}$$

$$R_{sh}^{min} = (\min(E_i, E_f) - CI_{end}) \times \min(CF)$$

where $CI_{end}$ is the confidence interval of the endpoints of the regression interval. Similarly,

$$R_{sch}^{be} = n.\, m. \times CF^{be}$$

$$R_{sch}^{max} = +n \times (m + CI_m)) \times \max(CF), \text{ if extraction is classified as I, CI, CD, or N,}$$

$$= n \times (m + CI_m) \times \min(CF), \text{ if extraction is classified as D;} \tag{5}$$

$$R_{sch}^{min} = n \times (m - CI_m) \times \min(CF), \text{ if extraction is classified as I, and}$$

$$= n \times (m - CI_m) \times \max(CF), \text{ if extraction is classified as D, CD, CI, or N.}$$

where the superscripts be, max, and min refer, respectively, to the best estimate, top of the range, and bottom of range; n is the number of years of the regression interval; m is the best estimate slope of the regression; and $CI_m$ is the confidence interval of the slope.

The impacts were classified as historically dominated if the minimum of the historical impact was greater than the maximum of the change impact, and vice-versa for impact-dominated. Otherwise, impacts were classified as neutral.

Time lags for impacts were estimated in two ways for the units with groundwater modelling outputs. The modelling reports provide the groundwater balance for the year 2016, the year 2060, and the period 2016–2060 for most scenarios. While there is a distribution of time lags for which impacts occur rather than a single value, the two methods estimated the time when a major component of the impact occurs. In the first approach,

$$t_l = (\Delta NAGD_{mean} - \Delta NAGD_{2016}) \times 54/(\Delta NAGD_{2060} - \Delta NAGD_{2016}), \qquad (4)$$

where $t_l$ is the estimate of the time lag and $\Delta NAGD_{mean}$ is the difference between mean stream groundwater exchange fluxes for the period 2016–2060 for two different scenarios, $\Delta NAGD_{2016}$ for 2016 and $\Delta NAGD_{2060}$ for 2060. Generally, Scenario 3 was chosen as the baseline scenario. By using a range of scenarios, a range of time lags was determined. The second approach used the modelling plots in the modelling reports to estimate the time lag at which most of the impact occurred. The figures vary across the reports and the impact may occur over extended periods of time.

*5. Assignment of risk*: the assignment of risks varies across the MDB. In the southern MDB with the large reservoirs and largely regulated streams, the risk may be expected to be less than the northern MDB with smaller reservoirs and more ephemeral and unregulated streams. Risk also depends on the environmental values being protected. The aim here was to provide metrics of risk as a basis for discussion. The NSW valleys, except the Murray, were used to exemplify the following indicators:

1. The ratio of maximum stream reduction from groundwater extraction to the lower baseflow thresholds. It was assumed that if the stream losses are comparable to this threshold, it means that it is increasingly difficult to supply these thresholds.
2. The ratio of maximum stream depletion to the surface water storage of that valley. It was assumed that if the ratio is high, the increased transmission losses along the regulated reaches lead to quicker depletion of the reservoir and greater difficulty in supplying the baseflow threshold. It also means that there is less capacity for the reservoirs to maintain baseflow for regulated reaches during dry periods, should transmission losses increase.
3. The ratio of maximum stream depletion to mean annual streamflow has been used previously [23] to indicate the low priority of the issue. A large ratio means that the ability of surface flow to maintain baseflow during dry times in the presence of groundwater extraction is reduced, especially for unregulated streams.
4. A further metric used in discussion of the issue has been the ratio of depletion to the volume of water recovery. The greatest volumes of recovered water have been in the southern MDB, where the greatest surface water diversion has occurred. The volume was designed to meet environmental targets around the MDB. The depletion by groundwater extraction greater than the recovered water raises issues of accountability of public funds used to recover water. It also should raise issues about whether the environmental targets will be reached.

The above metrics were defined for the maximum estimate of historical extraction, as mentioned in the previous step. This is a precautionary approach which accounts for the cumulative changes in extraction. The change in stream losses from the more recent trend was considered in the context of the historical impacts and hence only on those units

considered as moderate to high risk from the cumulative impact. Further considered in this context was the sensitivity of groundwater discharge to streams.

## 3. Results

*1. Collation of basin-wide datasets*: the temporal pattern of groundwater extraction for each of the eighteen priority units and the aggregate of these are shown in Figure 3. This shows that most of the units followed similar patterns with a minimum in the 2009-2011 period (years 6–8) and also in the 2016-2017 period (year 13). There was also an increase in extraction over the 2012-2013 to 2019-2020 period, except for a dip in the 2016-2017 period. While some units showed increasing trends with time, most had similar values for both the start and end of the time period.

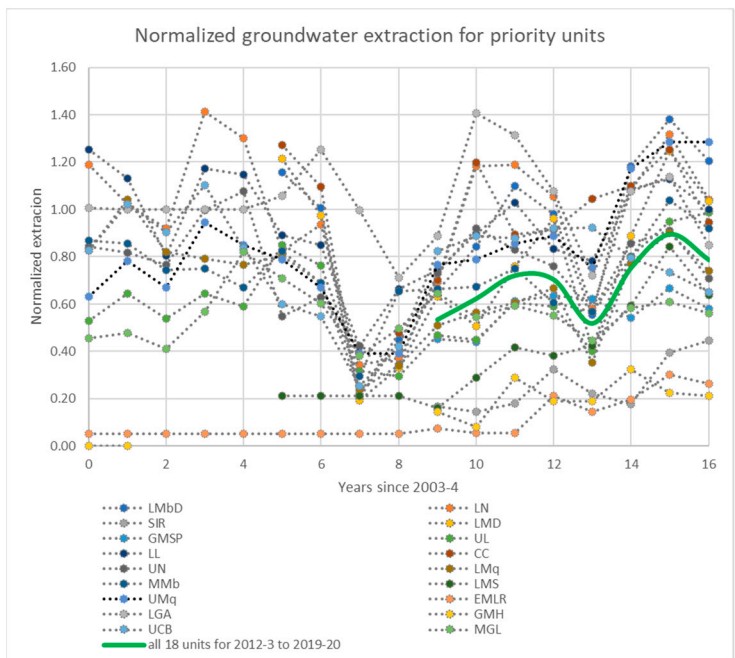

**Figure 3.** Annual groundwater extraction (GL/year) for the 18 priority groundwater units and for the aggregate of these, normalized with respect to the SDL, as annual time series for available data since 2003–2004. The continuous green line shows the result for the aggregate for the period of 2012–2013 to 2019–2020.

The scaled annual rainfall data are shown in Figure 4. The results showed a common pattern across the MDB with a wet period in 2009–2011 (years 6–8) and 2016–2017 and dry periods in years 0-6, 10-12 and 13-16. The SDL units occurred in four of the MDB states and provided a good geographic cover of the MDB. A visual comparison of the extraction and rainfall data would suggest a negative correlation with low extraction in wet year

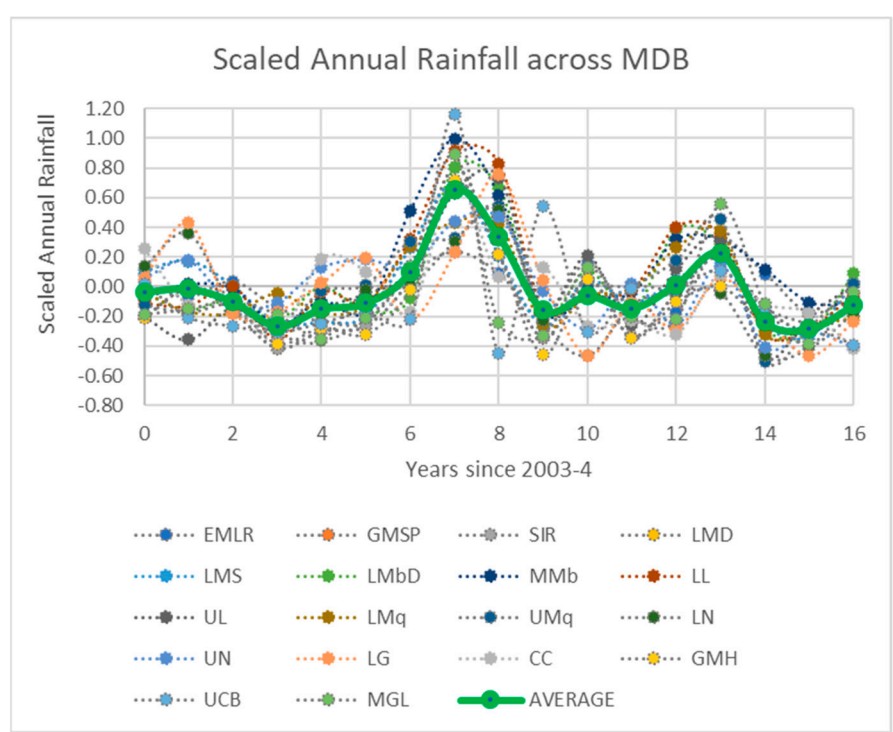

**Figure 4.** Scaled annual rainfall across the MDB and for each unit since 2003-2004.

*2. Temporal trends in groundwater extraction:* results of three different linear regressions are shown in Table A1 in Appendix A; namely, (1) normalized annual extraction with time over 2012–2013 to 2018–2019; (2) normalized annual extraction with time for entire time period defined by data availability; and (3) normalized annual extraction with time and scaled annual rainfall for entire time period of data availability. These were implemented for each of the priority SDL units, the aggregate of the priority units, and the aggregate of all eighty units.

The mean annual extraction for the period of 2012–2013 to 2019–2020 was 0.43 of the SDL, while that for the priority units was 0.69. The slope of the regression for all units was 2.0% SDL/year with a confidence interval of 1.9%/year; and for the priority units were 3.6 and 3.8, respectively. The trend with time and confidence interval for the multilinear regression (subsequently referred to as the rainfall-corrected trend) for all of the units were 1.7 and 1.4, respectively. If these trends continued for a 10-year Plan cycle, this would lead to an increase of about 20% of the SDL or nearly 700 GL/year. The data and regressions for normalized annual extraction are shown in Figure 5 in plots against (a) scaled rainfall and (b) years since 2003–2004. The plots in Figure 5a suggest a strong influence of rainfall with the initial two points lying below the trend line (indicating a slight trend despite the rainfall influence). However, the use of only eight datapoints for three parameters means difficulty in separating the long-term trend in extraction from short-term variations. Results in Table A1 in Appendix A show that the long-term trends for individual units over the period of 2003-2004 to 2019-2020 were very different to that of 2012-2013 to 2018-2019. Normally, about thirty years is required to define a reference period for climate. The main constraint to considering longer periods of time for the aggregate extraction is the availability of data in Victoria. Further analysis are therefore be constrained to the non-Victorian part of the MDB with the implications of this assumption being discussed later.

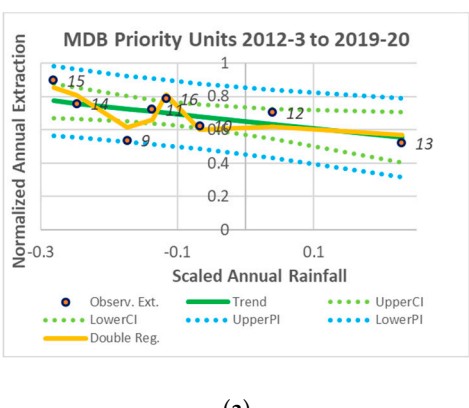 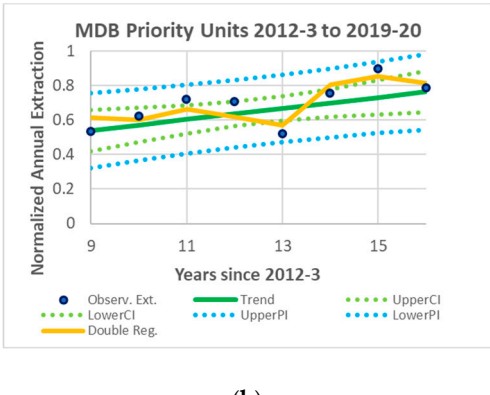

(**a**)                              (**b**)

**Figure 5.** Plots of normalized annual extraction for the whole of the MDB for the period of 2012–2013 to 2019–2020 (**a**) with respect to scaled annual rainfall and (**b**) with time (years since 2003–2004 year. For (**a**), trend line for double regression at year 2015 is also shown with confidence interval and prediction interval as well as the full double regression results. Data labels represent years since 2003–2004. For (**b**), the multilinear regression trend and full and linear regression for the 2012–2013 to 2018-19 period as well as the confidence and prediction intervals are shown.

Table A1 shows that there were three SDL units with increasing extraction (i.e., slope as estimated in the multiple regression is greater than the confidence interval), namely, LMS, UMq, and EMLR. One unit, UL, was considered conditionally increasing. The normalized annual extraction for the (a,b) Upper Lachlan and (c,d) eastern Mt Lofty Ranges' units are plotted in Figure 6 with confidence and prediction intervals against (a,c) scaled annual rainfall and (b,d) years since 2003–2004 and compared to the multiple regression. For the UL unit, the linear trend and rainfall-corrected trend for the whole period were almost identical, while the linear regression for the shorter period had a much greater slope and confidence interval. While the addition of scaled annual rainfall to the regression explained more of the variance, the prediction interval was still relatively high (0.34). While a slope was predicted, Figure 6a shows little evidence of the early years being below the trend line. High extraction volumes appeared to occur in the late drought period. Figure 6b indicates that extraction volumes could continue to increase for about ten years before the extraction limit is reached, but the uncertainty around this timing is high. The plot for EMLR Figure 6c,d show a groundwater unit in which extraction volumes have begun to increase in the time period, with more time required before the extraction limit is reached. The increasing trend is more evident with early years seen below the trend line in Figure 6c, while later years lie above the trend line.

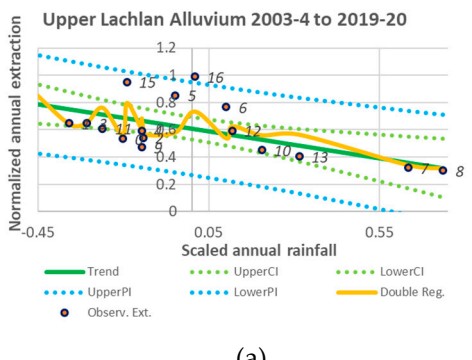 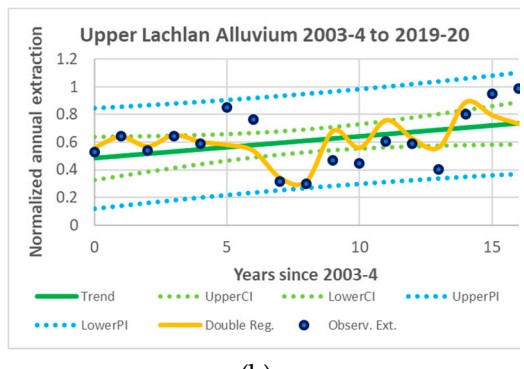

(a)                              (b)

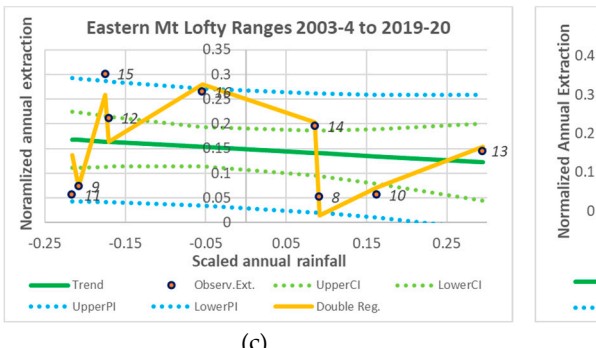
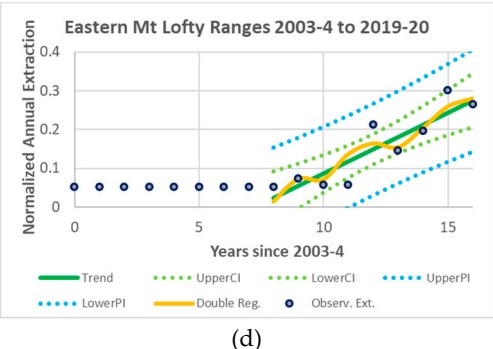

(c)                                                                                    (d)

**Figure 6.** Plots of the observed normalized annual extraction and multi-regression trend and full regression lines for the Upper Lachlan Alluvium, together with confidence and prediction intervals (**a**,**b**) and eastern Mt Lofty Ranges' (**c**,**d**) units for the period of 2003–2004 to 2019–2020. Plots (**a**) and (**c**) are against scaled annual rainfall, while (**b**) and (**c**) are against time (years since 2003–2004). Labels for (a) and (c) represent the years since 2003–2004.

Two units (LMD and UN) had decreasing slopes, while two units (LN and LMq) were conditionally decreasing. The rainfall-corrected trend for the LN was less than the linear regression and less than the slope from the linear regression for 2012–2013 to 2018–2019 (3.8 (13.9)), although error bands for all overlapped. While the addition of scaled annual rainfall to the regression explained more of the variance, the confidence and prediction intervals were still relatively large (0.15, (0.6)). This reflects the assumed linear relationship between extraction and rainfall being imperfect. All four units were part of the ASGE and subject to reductions in entitlements and extraction limits, with the LMD showing the most obvious decline.

The extraction in the remaining eight units (LMbD, LL, CC, MMb, LG, UCB, and MGL) were classified as neutral. The trends for the LMbD showed that the linear regression and rainfall-corrected trend for 2003–2004 to 2019–2020 were similar but much lower than the regression for 2012–2013 to 2018–2019 (8.2 (12.4)). While the addition of scaled annual rainfall to the regression explained more of the variance, the confidence interval remained relatively high (0.21).

The results from the individual non-Victorian priority units were aggregated to form an aggregate trend for these units. More specifically, the linear and rainfall-corrected trends and multiple regression were used. The results for the rainfall-corrected trend are shown in Figure 7 and compared to observed data and the multiple regression from using that data directly. The slope of the aggregated linear trend (−0.3 %SDL/year) was close to that of the aggregated rainfall-corrected trend (−0.7 %SDL/year) but much smaller than that of the rainfall-corrected trend of the measured data (0.9 (3.7) %SDL/year) for the linear regression over the 2012–2013 to 2018–2019 period (4.5 (7.2) %SDL/year). However, all these estimates lay within the confidence interval shown in Figure 7b. These results would indicate that the slope of the long-term trend of the aggregate extraction is not significantly different from zero. The average extraction was about 960 GL/year with a confidence interval of 130 GL/year and predictive interval of 460 GL/year.

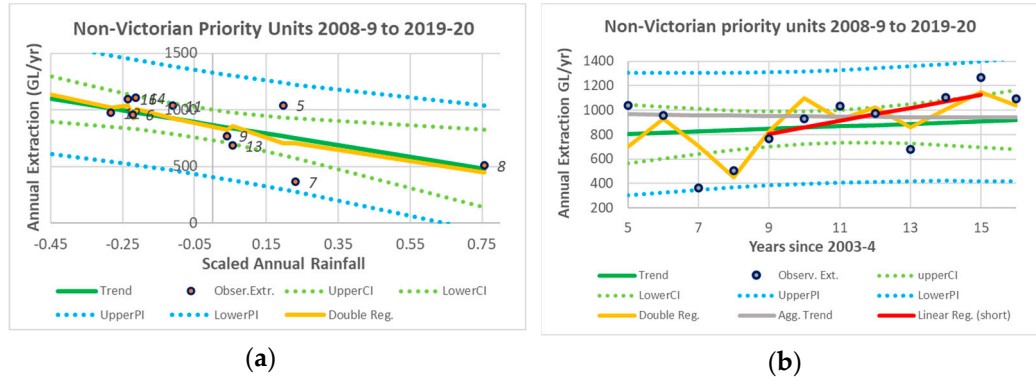

**Figure 7.** Plots against (**a**) scaled annual rainfall and (**b**) years since 2003–2004 (**a**) represent the years since 2003–2004 of the normalized annual extraction (GL/yr) for the non-Victorian priority units over the period of 2003–2004 to 2019–2020. The observed data (from 2008–2009) were compared to the aggregate of the regression trend from individual units (from 2003–2004) and the multiple regression trend and full regression lines for 2008–2009 to 2019–2020. The linear regression for the 2012–2013 to 2018*2019 period is also shown. The labels in (**a**) represent the years since 2003–2004.

The results from the non-Victorian priority units indicate that the trend slope reduced as the length of the regression period increased. There were two different effects. The first was that the correlation between rainfall and time diminished with length of the time period. Short periods led to the extraction being sensitive to short-term trends in rainfall. The trend in rainfall is equivalent to the slope of a moving average of the rainfall. Secondly, the longer period provided more datapoints on which to base statistics. Figure 8 shows that the pattern of extraction trends broadly matched that of rainfall trends as the period increased. This suggests that about twenty years are needed to obtain a more stable estimate of the decadal trend. This supports the decision not to include the Victorian units for further analysis as data were readily available for only 8 years.

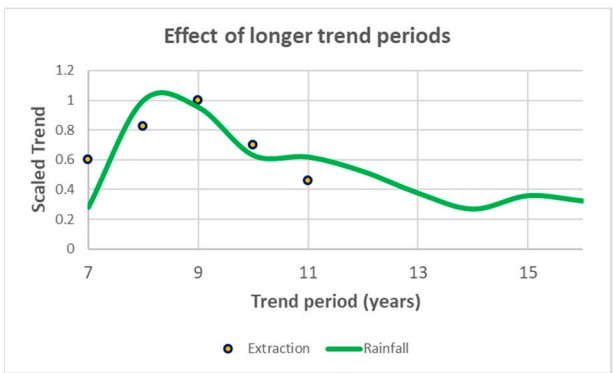

**Figure 8.** Plot showing the effect of longer time periods on trend estimates for both annual rainfall and extraction. Estimates of slope were obtained by least-squares linear regression of rainfall or extraction against time for a period up to 2019–2020. The *x*-axis represents the start of the trend period in the number of years before 2019–2020. Both trend estimates were normalized with respect to the maximum value.

3. *Connectivity and time scale estimates*: Table 3 shows the results of the estimation of CF. In ten of the fourteen alluvial units, the linear regression showed a very high correlation ($R^2 > 85\%$). One unit (LG) indicated a moderate correlation (63%) and two (CC, LMq) had low correlation (<12%). Both those with low correlation had a very low estimate for CF (<0.07). For the Lower Lachlan, CF determined from the recharge scenarios was different from that determined from extraction scenarios, but both were very low. The results showed that CF for four units (LMbD, LL, CC, and LMq) were very low (<0.1), two

(LN and LG) were low (0.1 < CF < 0.3), six (SIR, LMD, GMSP, UL, MMb, and LMS) were moderate (0.3 < CF < 0.7), and two (UN and UMq) were high (CF > 0.7). All the units with very low and low values were riverine plain alluvial groundwater systems, while those with high values were broad-valley-constrained floodplain alluvia. Those with moderate values were a mixture of both.

**Table 3.** Estimates of CF for each groundwater unit as determined from groundwater modelling output, together with the confidence interval and $R^2$ from the regression analysis; the assigned lower and upper limits for the range of CF; the ratio of discharge to the total stressor; sensitivity of discharge to stressor; and the time lag calculated by Equation (6) (time1) and by visual inspection (time2).

| SDL Unit | CF | CI (CF) | $R^2$ | $CF_{min}$ | $CF_{max}$ | Disch. Ratio | S Disch. Ext. | time1 years | time2 years |
|---|---|---|---|---|---|---|---|---|---|
| LMbD | 0.06 | 0.0 | 99 | 0 | 0.15 | 0.10 | 1.7 | 10-25 | 10 |
| LN | 0.11 | 0.05 | 89 | 0.05 | 0.3 | 0.1 | 1.0 | 10-20 | 10 |
| SIR | 0.38 | 0.03 | 99 | 0.2 | 0.6 | 0.32 | 0.87 | 15-25 | 10 |
| LMD | 0.42 | 0.02 | 99 | 0.2 | 0.6 | 0.67 | 1.47 | 10-25 | 10 |
| GMSP | 0.38 | 0.03 | 99 | 0.2 | 0.6 | 0.32 | 0.87 | 15-25 | 10 |
| UL | 0.43 | 0.02 | 96 | 0.2 | 0.6 | 0.02 | 1.75 | 10-20 | 10-20 |
| LL | 0.01 | N/A | N/A | 0 | 0.15 | 0.46 | 0.20 | NA | 10 |
| CC | 0.01 | 0.04 | 10 | 0 | 0.15 | 0 | 0 | 10-20 | 10-15 |
| UN | 0.81 | 0.02 | 99 | 0.5 | 1 | 0.17 | 2.73 | 7-15 | 10-25 |
| LMq | 0.02 | 0.02 | 9 | 0 | 0.15 | 0.28 | 6.65 | NA | 10 |
| MMb | 0.51 | 0.0 | 100 | 0.3 | 0.7 | 0.17 | 0.92 | 8-12 | 10-25 |
| LMS | 0.42 | 0.01 | 99 | 0.2 | 0.6 | 0.67 | 1.47 | 10-25 | 10 |
| UMq | 0.82 | 0.17 | 96 | 0.5 | 1 | 0.38 | 0.43 | NA | 5-25 |
| EMLR | 0.75 | NA | NA | 0.5 | 1.0 | NA | NA | NA | NA |
| LG | 0.21 | 0.2 | 63 | 0.05 | 0.5 | 0.18 | 0.37 | 15-30 | 10 |
| GMH | 0.75 | NA | NA | 0.5 | 1 | 1 | 0 | 0-10 | 0-10 |
| UCB | 0.75 | NA | NA | 0.5 | 1 | 1 | 0 | 0-10 | 0-10 |
| MGL | 0 | NA | NA | 0 | 0.01 | 1 | 0 | >100 | >100 |

For the modelled units, the captured discharge was only higher than the induced recharge for the Lower Murray units. The discharge ratio was low (disch. ratio < 0.3) for seven units (LMbD, LN, UL, UN, LMq, and LG) and moderate (0.3< disch. ratio <0.5) for four units (SIR, GMSP, LL, and UMq). The discharge was sensitive to extraction (S dich. Ext. >1) for seven units (LMbD, LN, LMD, UL, UN, LMq, and LMS). The discharge to streams in the westerly part of the southern Riverine Plain model that modelled the LM, SIR, and GMSP units used a drainage module to simulate discharge to streams, and some caution needs to be applied in using these parameters. It also should be noted that the deep aquifers and shallow aquifers for each state (LLMD and LMS for NSW; SIR and GMSP for Victoria) should be considered together. This analysis showed that induced recharge dominated captured discharge for the alluvial groundwater systems of the MDB. The sensitivity seems to be unrelated to whether groundwater systems are riverine plain or mid-valley systems, but it requires the ratio of stream inflow to extraction to be small. The largest sensitivity was for the Lower Macquarie, where CF was very small and the ratio of discharge to streams relative to extraction was less than 0.001.

The delay between action and response appeared to be about ten years in general, with some slightly shorter responses for the mid-valley units (UN, MMb, and UMq) and larger response times for the Goulburn-Murray units (SIR and GMSP). The adoption of a ten-year response time in planning means that the measured increases in extraction rates

over a ten-year period can be used in the next planning cycle, potentially simplifying the process for adaptation. The long lag time means that avoidance based on monitoring is difficult, while offsets can be pursued with more certainty.

*4. Impacts on streamflow*: Table 4 shows the impacts of extraction for non-Victorian units. The best estimate of the impact from historical extraction was -300 GL/year, with a potential range of -560--125 GL/year. The best estimate of the impact of changed extraction was -44 GL/year with a potential range of -185-105 GL/year. The impacts were rising for three units (LMS, UMq, and EMLR), conditionally increasing for UL, neutral for seven units (LMbD, LL, CC, MMb, LG, UCB, and MGL), conditionally reducing for two units (LN and LMq), and reducing for two units (LMD and UN). The average CF for the best estimate impact was ~0.3. Two of the units had impacts which were dominated by the change in extraction (LMS and EMLR), two units had neutral impacts (UL and UMq), and the remaining eleven units were dominated by historical extraction. Four units dominated the magnitude of the historical impact (UN, LMD, UL, and MMb), while LMD and UN dominated the impact from changed extraction by causing an overall reduction.

**Table 4.** Impacts of extraction for non-Victorian units, showing best estimate and range of impact from historical extraction and for the change in extraction. In the column 'Increasing?', N denotes neutral, I increasing, CI conditionally decreasing, D increasing, and CD conditionally decreasing, and the value in brackets is the *p*-value associated with the rate of increase; while under the column dominance, N denotes neutral, hist denotes impact dominated by historical extraction, and change denotes impact dominated by the change in extraction.

| SDL Unit | Best Estimate Hist. Stream Reduction (GL/year) | Maximum Hist. Stream Reduction (GL/year) | Minimum Range Hist. Stream Reduction (GL/year) | Increasing? (*p*-value) | Best Estimate Change Stream Reduction (GL/year) | Maximum Range Change Stream Reduction (GL/year) | Minimum Range Change Stream Reduction | Dominance |
|---|---|---|---|---|---|---|---|---|
| LMbD | 17 | 531 | 0 | N (0.89) | 0 | 15 | 0 | hist. |
| LN | 10 | 34 | 4 | CD (0.085) | −3 | 2 | −3 | hist. |
| LMD | 49 | 85 | 18 | D (0.011) | −26 | −3 | −64 | hist. |
| UL | 30 | 50 | 11 | CI (0.059) | 10 | 30 | −1 | neutral |
| LL | 1 | 20 | 0 | N (0.42) | 0 | 3 | 0 | hist. |
| CC | 0 | 9 | 0 | N (0.74) | 0 | 6 | 0 | hist. |
| UN | 91 | 131 | 47 | D (0.017) | −34 | −4 | −75 | hist. |
| LMq | 1 | 7 | 0 | CD (0.09) | 0 | 0 | 0 | hist. |
| MMb | 20 | 34 | 9 | N (0.88) | 0 | 5 | −10 | hist. |
| LMS | 1 | 1 | 0 | I (0.002) | 0 | 2 | 0 | change |
| UMq | 16 | 22 | 9 | I (0.002) | 7 | 13 | 3 | neutral |
| EMLR | 8 | 13 | 4 | I (0.002) | 7 | 14 | 3 | change |
| LG | 8 | 20 | 0 | N (0.19) | −1 | 2 | 0 | hist. |
| UCB | 48 | 81 | 24 | N (0.55) | −6 | 20 | −35 | hist. |
| MGL | 0 | 0 | 0 | N (0.47) | 0 | 0 | 0 | hist. |
| Total (15 units) | 299 | 558 | 125 | I (0.034) | −44 | 104 | −183 | |

*5. Risk assignment*: Table 5 shows the risk indicators for five NSW river valleys. The Namoi valley had the highest ratio relative to surface water availability (17%), followed by the Lachlan (6%). These river valleys also had the highest impacts relative to surface water storage corresponding to 5–16 years. The Namoi had, by far, the greatest impact relative to water recovery (15.7), followed by the Lachlan valley (1.5). Finally, the Namoi and Lachlan had the greatest impact relative to baseflow thresholds (3.0 and 1.7, respectively, for lower threshold and 1.3 and 0.7, respectively, for upper threshold).

Within the Namoi and Lachlan valleys, the impact from extraction was decreasing or conditionally decreasing for the UN and LN, neutral for LL, and conditionally increasing for UL. UN dominated the historical impact in the Namoi, which dominated the impact from changed extraction. UL dominated the impact in the Lachlan and was neutral with respect to change and historically dominated change. The best estimate impact from historical extraction was 40% lower than the maximum estimate in the Namoi and 60% lower in the Lachlan, and while these would lead to lower values of the risk indicators, each would still be significant. The maximum estimate of the impact since 2003 was -1.3% of that from the historical pattern of extraction for the Namoi, while for the Lachlan, it represented 46%. These would have the impact of slightly improving the risk in the Namoi but significantly exacerbating the risk in the Lachlan.

**Table 5.** Risk indicators for each valley not including Victorian or South Australian rivers. The maximum aggregate impact of priority units is shown as well as the ratio of this to annual surface water availability [21], surface water storage [89], surface water recovery targets [90], and baseflow thresholds [91–96] as determined by most relevant gauging station in that valley.

| River Valley | Maximum Reduction Streamflow (GL/year) | Ratio of Maximum Reduction in Streamflow to | | | | | Indicator Station for Baseflow |
| | | Annual Surface Water Availability (%) | Surface Water Storage (year⁻¹) | Water Recovery | Lower Baseflow Threshold | Maximum Baseflow Threshold | |
|---|---|---|---|---|---|---|---|
| Murrumbidgee | 84 | 1.96 | 0.03 | 0.19 | 0.23 | 0.06 | 410,005 |
| Lachlan | 70 | 6.17 | 0.06 | 1.49 | 1.68 | 0.69 | 412,038 |
| Macquarie-Castlereagh | 29 | 1.84 | 0.02 | 0.30 | 0.40 | 0.40 | 421,004 |
| Namoi | 165 | 17.12 | 0.18 | 15.74 | 3.02 | 1.29 | 419,012 |
| Gwydir | 20 | 2.60 | 0.01 | 0.37 | 1.11 | 0.22 | 418,053 |

## 4. Discussion

This paper developed and tested an experimental reporting approach to support management of the impacts of increased groundwater extraction on baseflow in the MDB. In this section, we discuss each component of the approach and then the approach as a whole.

*Prioritizing groundwater units*: the prioritization is seen as a critical component of the approach in allowing sufficient focus on those units, where groundwater extraction is occurring. In this case, fourteen of the units were alluvial groundwater systems with groundwater models and a history of investigations. The approach also identified units where groundwater extraction was only beginning to increase. This suggests that the approach works as an early warning system, allowing time for investigations, before issues arise. If the approach was to be adopted, the time requirements for units already identified would reduce in future implementations, allowing more time on units with emerging groundwater extraction increases. Changing the number of units would affect the balance between focusing resources and detecting emerging risks.

*Trend analysis*: the trend analyses aimed to provide long-term trends for priority units and for the MDB. The non-Victorian component of the MDB showed no significant trend for the period of 2003–2004 to 2019–2020. The time series of available data for Victorian units was not sufficiently long to provide confidence in trends. However, these datasets could be developed readily. From preliminary analyses of available data, trends are not expected to be very different for Victoria than for the rest of the MDB. The analyses showed that about twenty years of annual data are required to develop trends with confidence. This means that analyses should improve in future implementations. As more data become available, it should be feasible to develop more sophisticated trend approaches, such as a more flexible approach using spline fitting to demonstrate any changing trends.

While rainfall explained much of the interannual variability for individual units and the Basin, regressions using rainfall could not reduce the need for longer datasets. There are many potential causes for remaining variability, including nonlinear relations with historical rainfall and changes in use associated with groundwater trade, drought, changing surface water limits, implementation of groundwater plans, changing commodities, and carry-over allowances. Understanding the causes of variability is important for supporting management and policy and predicting future trends. Such relationships have been used for surface water for some time, most notably to ensure compliance. However, the regressions explained enough of the variance to be confident that the trends from 2012–2013 to 2018–2019 were due to processes with short-term variability of rainfall.

There was no consistent trend across priority units. The annual extraction was stable at about the SDL for four units (LMbD, LL, CC, and LG); decreased in four units from the SDL (LN, LMD, UN, and LMq); stabilized below the SDL for three units (MMb, UCB, and MGL), increased to the SDL in one (UMq), and was increasing but not yet at the SDL for three units (UN, LMS, and EMLR). All the units with decreasing trends were part of the ASGE scheme and with transitional supplementary licenses. The increasing trends may be expected to result from long-term drought conditions.

Reasons for extraction not increasing for units outside the priority units include groundwater salinity, low aquifer transmissivity, and lack of demand. However, these do not appear to apply to many of the units for which there were no increases. The average extraction in NSW alluvial systems for 2012–2013 to 2018–2019 was about 72% of their aggregate SDL. The priority units did not include any highly connected units except a minor part of the MMb. It is unclear why the recent drought led to increases in shallow alluvial groundwater systems (e.g., LMS) or fractured rock systems (e.g., EMLR), yet not in these alluvial systems. Until this is better understood, it would be difficult to be confident about extraction not increasing across the MDB for the near future. It is beyond the scope of this study to explain all the trends or lack of trends beyond some speculation.

*Connectivity*: this component provided an estimated range for CF and the associated discharge parameters for priority units. The use of the general CF approach within a risk framework allowed estimates for CF to be changed incrementally as better estimates became available and to be flexible about the scenarios to be considered.

As most of the units were alluvial, groundwater modelling outputs were used to provide estimates for those units. There are some issues with the groundwater models used for this purpose, but they currently provide the best possible estimate until the models are improved and more relevant scenarios become available for use. The robustness of the value of CF across various scenarios showed that CF can be used to interpolate modelling results from these scenarios as long as the scenario of interest does not change significantly. Better scenarios (calibration, scenario 1/2) could have been chosen for the historical impact, and this may lead to an under-estimate of impact for some units. As extraction continues to increase, more of the unit becomes disconnected with the stream system and the maximum impact is reached, implying that CF becomes zero. The uncertainty in CF due to modelling can be tested through inter-comparison of models. For example, another groundwater model was developed in 2012 for the Upper Lachlan alluvium [96], encapsulating the whole groundwater unit and having a later date of development. The estimate of CF from this model fell almost within the uncertainty range assigned to this unit. The value of 0.16 for the transition from the current development to full SDL is lower than the best estimate in this paper. This would be affected by the lower connectivity of the unregulated reaches added to the unit in the more recent model and by the deeper water tables over recent times. As expected, CF was generally lower for the riverine plain alluvial groundwater systems than for the broad-valley-constrained floodplains.

The adopted range for fractured rock (0.5–1.0) reflects the difficulty of fractured rock systems. These units are often large and comprise several streams, different geologies, local flow systems, and localized pumping, and are sensitive to fracture patterns. While this is unsatisfactory, the impacts are likely to be local rather than contributing to regional impacts, and therefore, a finer resolution approach is needed to improve estimates.

Apart from the Lower Murray, the captured discharge was less than the induced recharge. This partly reflects the state of the groundwater system where adjacent streams have shifted towards losing streams as well as the siting of groundwater extraction near losing parts of the streams. This means that salinity improvements from extraction may not be as great as previously thought. While both induced recharge and captured discharge can affect stream salinity, the high salinity of groundwater often found in

Australia means that captured discharge generally has a higher impact. More importantly, the discharge to streams may be important to protect refuges in the stream environment.

The moderate sensitivities of discharge to extraction mostly reflects the small discharge to extraction ratio. Of those units with increasing trends, UL is the unit of most interest. It had both a low captured discharge to change in streamflow (0.02) and a low stream discharge to extraction ratio (0.004). The moderate sensitivity means that increased extraction over the last twenty years would have significantly affected the low volume of discharge into streams. Of the units where extraction was decreasing, UN and LN are of most interest. The modelled ratio of groundwater discharge to streams relative to extraction was not as low as for the UL, and so the moderate sensitivity indicated that any significant reduction in extraction would lead to both a moderate absolute change in groundwater inflow as well as a moderate relative decrease.

The time lags between changes in extraction and streamflow were very significant. The groundwater hydrographs for the Namoi and Lachlan appeared to be indicating that the groundwater systems are still responding to the historical increase in extraction from twenty years ago, perhaps exacerbated by the drier climate over that time. The large time lags mean that offsets, e.g., releases from surface water storage, can be considered in the following plan cycle but makes avoidance based on measured responses difficult. Trends in low flows are difficult to measure because of the large stream variability, but it is possible [35,97]. Relating patterns in flow to groundwater-related causes are made more difficult because of these lags.

*Impacts*: the impact of increased groundwater extraction on streamflow was shown to be not increasing for the non-Victorian component of the MDB. While there was significant uncertainty, most units showed decreasing or neutral impacts. There were four units for which the impact was increasing or conditionally increasing, implying that any impact would be regional or local rather than basin-wide.

The impacts also showed that historically established patterns of extraction dominated changes that have been occurring over the last twenty years. The best estimate of the impact of historically established patterns of extraction from priority units was ~300 Gl/year, while there has been possibly a negative change in impact from extraction due to changes in extraction over the last twenty years. The extraction rate corresponds to a mean value of CF of ~0.3.

*Risk Indicators*: the risk indicators were developed for five NSW valleys, which suggested the Namoi and Lachlan were the valleys of highest risk. The use of the best estimates of impact rather than maximum values may lower the metrics of risk but not the selection of these valleys. The high-risk metrics are consistent with groundwater behaviour of the regional groundwater systems of the Namoi and Lachlan River valleys. These systems have falling water tables that are still responding to extraction patterns established twenty years ago [99–101]. In several management zones, water tables have fallen below 25% of the total available drawdown (TAD) and are approaching this in others. Much of the UN and part of the LN are still connected to the Naomi River and tributaries [100], while the UL is connected to the Lachlan River [51], and the LN and LL are mostly disconnected [51,102–105]. The zones of disconnection have been growing in the Namoi [100], and there have been several studies in the Namoi showing shifts from gaining streams to losing streams [106–114].

The metrics are also consistent with estimates of stream depletion. Groundwater models show a depletion of 38.4 GL/yr during development [82]; Upper Namoi, 63.8 GL/yr [88]; Lower Lachlan, 0.34 GL/yr [79]; and 18.1–33 GL/yr {86, 96] for the Upper Lachlan. There are also studies of declining streamflow in the Namoi [35,97]. The groundwater and stream monitoring are critical for the assessment of these indicators; but importantly, towards developing management to protect groundwater–surface water connectivity. The temporal variability of streamflow and its sensitivity to stream management means that it is difficult to use directly for adaptive management.

Groundwater levels and surface water storage levels may provide an important surrogate for the ability to deliver baseflows.

The metrics are consistent with the difficulty of delivering environmental flows during the recent 2017–2020 drought [115,116]. The Keepit and Split Rock Dams in the Namoi had minimal water in storage, while levels in the Wyangala Dam in the Lachlan fell to 8%. Environmental flows were restricted over the period but allowed for some critical watering. Fish rescues were required in both valleys, with many reports of fish deaths [115,116].

Environmental water plans highlight risk with high-connectivity alluvial systems. The quick-responding high-connectivity zones are subject to conjunctive water management rules [94] that define cease-to-pump rules for unregulated near-perennial streams to protect perennial pools in the streams or link annual water determination for groundwater with that for surface water in regulated streams to control stream losses. The analysis in this paper did not feature these systems, as the annual changes in extraction volumes for these units have been small.

However, the environmental watering plans assume that compliance with the SDL should mitigate any groundwater extraction risk [92,94]. The SDLs have been determined so that, in conjunction with groundwater management plans, there is an acceptable level of impact on streams [117]. The SDL should be considered in conjunction with other plan rules, such as the recovered water table threshold of 25% TAD. Thus, while groundwater extraction may have contributed to the risk with baseflows, its management is not currently seen as part of mitigating the risk. The difficulty of delivering required baseflows may mean that this needs to change. Managing water levels to higher trigger levels may be required to recover some of the baseflow lost to groundwater depletion without changing SDLs. There may be some opportunity for managed aquifer recharge to support such management.

While the value of the risk indicators is largely based around pre-2003 patterns, more recent trends in extraction in the UL and UM have increased the risk values, exacerbating the effects of prior extraction patterns. Of these two, that of the UL is more concerning because of the context in which it is occurring. Increases in extraction since 2003 is predicted to cause almost 50% of the impact from historical patterns of extraction. Extraction could still increase further from its current situation of about 75% of the SDL. The two units, where increased extraction is beginning to emerge, have not been highlighted as part of the risk indicators. EMLR is a complex fractured rock Tertiary limestone connected to more than one stream and would need to be considered in finer detail. The current indicators have not addressed unregulated streams outside of the highly connected areas. Apart from cease-to-pump rules for both surface and groundwater, there appears to be little opportunity to support baseflow in these systems. Better groundwater management may be able to support high-priority systems, with the flow duration curve being used to devise management targets. The ad hoc nature of these systems means that such targets and risk indicators would need to be localized. The LMS is connected to a highly regulated river system with large river storages and with major water and land use changes occurring. An increase in extraction is unlikely to be significant relative to these other processes. While trends in extraction were not significant for non-priority units, their pre-2003 extraction affects the risk values. Ideally, these should be included in the risk analysis, but the priority units capture the majority of the pre-2003 levels of extraction.

The definitions of these risk indicators have benefited from much clearer environmental management objectives for baseflow. Values have been partially incorporated into the definition of the thresholds and targets; but inevitably, there will be a need to balance the reality of climate variability and economic and social values with consumptive use and environmental values. The need for fish rescues in the Namoi and

Lachlan has highlighted the change in community values, especially following major deaths on the Darling River.

*Reporting framework as a whole*: the components of the framework achieved most of the intended aims. The final output did not cover the majority of the MDB for various reasons. The time series was not sufficiently long in Victoria to have a confident trend estimate, and other areas could have benefitted from a longer series. The connectivity, impact, and risk analyses were problematic for fractured rock areas due to localized conditions and environmental assets. However, the SDLs for these areas were set at a low percentage of run-off, and any impact should be much smaller than that of a drier climate [115,116,117]. The NSW Murray was not included in the risk assessment as the southern connected river system, which is also adjacent to the Victorian groundwater units, is managed as an integrated system.

The lack of a trend means that most of the imminent risk outside of the Lachlan occurs from historical increases in extraction rather than from current trends. However, should this change, the reporting system demonstrated that it can identify units, which are just beginning to emerge as a threat. The annual extraction volumes of the EMLR and LMS units have both begun to increase in recent times. The framework provides an approach to assessing such risks.

There is a need for the reporting system to be conducted regularly to support adaptive management. There is not much benefit in the analysis being conducted on an interval of less than five years as trends are unlikely to be changed. It also cannot be extended to beyond ten years, given that extraction patterns could change more quickly from changes in policy, economic conditions, climate, or technology. Any reporting has to line up with other reviews and reporting should allow for that 5–10-year cycle.

The applicability of the approach to other basins is dependent on administrative arrangements and availability of information. As these will vary from basin to basin, it is unlikely that the detailed approach would be transferrable to other areas. However, the framework is generic and may be transferrable, although detailed methodology will need to be adapted. The framework was most relevant to alluvial systems connected to regulated rivers.

## 5. Conclusions

The paper describes an approach to reporting the impacts of increasing extraction on baseflow. The aim was to highlight those units where the changing extraction may be a risk to delivering environmental baseflows. The reporting framework consists of five components: (1) initial prioritization of units; (2) trend analysis; (3) connectivity estimates; (4) impact assessment; and (5) assignment of risk.

Eighteen groundwater units, fourteen of which are alluvial, were identified from eighty units to have had significant changes in extraction over the 2012–2013 to 2018–2019 period and to have been responsible for 95% of the 53% increase that occurred in this period. Of these eighteen units, three Victorian units had insufficient data for confident trend analysis, four had increasing trends in extraction (and impact), and four had decreasing trends. The connectivity factor and associated properties were estimated using groundwater modelling for the alluvial systems and assigned for the remaining systems. The aggregate trend in both extraction and impact for the component of the MDB outside of Victoria had no significant trend, implying that any risk from increased extraction is either local or regional rather than basin-wide. Of the river valleys in New South Wales (excluding Murray and Border Rivers), historical extraction in the Namoi and Lachlan had the most risk to maintaining environmental baseflows, with this being exacerbated by recent trends in the Upper Lachlan.

Implementation to the whole of the MDB should be possible with some caveats. The approach worked much better for alluvial groundwater systems connected to regulated rivers than for the fractured rock areas or unregulated streams. Records of twenty years

are required for confident trend analysis. Clear targets are required for baseflow at appropriate reaches of the river. Where these are met, the method identifies those areas where risk is highest. If implemented, a reporting cycle of five to ten years would appear to be appropriate to align with planning cycle, with the required resources reducing with each iteration.

**Funding:** This research received no external funding.

**Institutional Review Board Statement:** Not applicable.

**Informed Consent Statement:** Not applicable.

**Data Availability Statement:** Not applicable.

**Conflicts of Interest:** The author declares no conflict of interest.

## Appendix A

**Table A1.** Showing the starting period for data availability, mean for the period of 2012–2013 to 2018–2019 and for the whole period of data availability, slope with standard error for the period of 2012–2013 to 2018–2019 and for the whole period of data availability; the multiple regression statistics (slope with standard error and $R^2$) and the predictions at the initial values and final values of data availability and the standard error in the predictor. In column 2, * denotes that extraction for deep aquifer has been derived from total extraction for groundwater unit by using an assumed value for extraction from shallow unit before 2007-8.

| ID | Time Period (Start) | Mean (2012–2013 to 2018–2019) | Mean (Total Period) | Linear Regression Slope (2012–2013 to 2018–2019) % SDL/year | Confidence Interval Slope % SDL/year | Linear Slope (Total Period) % SDL/year | Confidence Interval (Total Period) | Mult. Reg. Slope % SDL/year | Confidence Interval Slope % SDL/year | $R^2$ | Initial Value of Regression Line | Final Value of Regression line | Confidence Interval |
|---|---|---|---|---|---|---|---|---|---|---|---|---|---|
| LMbD | 2003–2004 * | 0.99 | 0.97 | 8.2 | 12.4 | −0.3 | 3.6 | 0.1 | 2.2 | 64.6 | 0.99 | 1.01 | 0.21 |
| LN | 2003–2004 | 1.01 | 0.97 | 3.8 | 13.9 | −0.3 | 3.4 | −2.2 | 2.6 | 54.1 | 1.08 | 0.71 | 0.22 |
| SIR | 2012–2013 | 0.23 | 0.26 | 2.8 | 3.8 | 3.7 | 3.0 | 3.7 | 3.3 | 63.0 | 0.12 | 0.38 | 0.16 |
| LMD | 2003–2004 * | 0.81 | 0.94 | 8.0 | 12.1 | −3.5 | 4.1 | −4.3 | 3.2 | 56.4 | 1.30 | 0.60 | 0.3 |
| GMSP | 2012–2013 | 0.57 | 0.57 | 3.1 | 3.2 | 2.1 | 2.7 | 2.2 | 3.1 | 39.4 | 0.49 | 0.64 | 0.14 |
| UL | 2003–2004 | 0.61 | 0.61 | 6.9 | 7.1 | 1.2 | 2.1 | 1.6 | 1.7 | 49.8 | 0.48 | 0.74 | 0.15 |
| LL | 2003–2004 | 0.93 | 0.93 | 4.6 | 6.2 | −0.7 | 2.3 | −0.5 | 1.4 | 66.9 | 1.01 | 0.92 | 0.13 |
| CC | 2008–2009 | 1.01 | 0.93 | 5.7 | 7.8 | 2.5 | 5.9 | −1.0 | 6.0 | 43.9 | 0.98 | 0.84 | 0.35 |
| UN | 2008–2009 | 0.80 | 0.75 | 0.5 | 6.5 | −0.3 | 2.1 | −2.1 | 1.7 | 58.7 | 0.91 | 0.57 | 0.15 |
| LMq | 2003–2004 | 0.63 | 0.67 | 4.8 | 7.7 | −1.3 | 2.2 | −1.2 | 1.5 | 61.5 | 0.77 | 0.57 | 0.14 |

| | | | | | | | | | | | | | |
|---|---|---|---|---|---|---|---|---|---|---|---|---|---|
| MMb | 2003–2004 | 0.73 | 0.73 | 4.3 | 6.9 | 0.2 | 1.8 | 0.1 | 1.7 | 12.9 | 0.72 | 0.74 | 0.16 |
| LMS | 2003–2004 * | 0.04 | 0.07 | 1.6 | 0.7 | 1.4 | 0.5 | 1.2 | 0.5 | 86.5 | 0.03 | 0.12 | 0.02 |
| UMq | 2003–2004 | 0.93 | 0.82 | 8.0 | 6.6 | 2.9 | 2.3 | 3.1 | 1.3 | 78.2 | 0.57 | 1.08 | 0.13 |
| EMLR | 2003–2004 | 0.15 | 0.10 | 3.7 | 2.4 | 3.2 | 1.4 | 3.1 | 1.4 | 83.3 | 0.05 | 0.27 | 0.07 |
| LG | 2003–2004 | 1.08 | 1.02 | −1.8 | 11.2 | 0.0 | 2.0 | −1.1 | 1.8 | 43.9 | 1.10 | 1.02 | 0.13 |
| GMH | 2012–2013 | 0.21 | 0.21 | 2.3 | 3.6 | 1.6 | 2.7 | 2.5 | 2.7 | 56.1 | 0.08 | 0.26 | 0.13 |
| UCB | 2003–2004 | 0.85 | 0.77 | −1.5 | 3.3 | −0.6 | 2.3 | −0.6 | 2.2 | 16.1 | 0.81 | 0.71 | 0.21 |
| MGL | 2003–2004 | 0.57 | 0.56 | −0.6 | 3.3 | 0.2 | 1.1 | 0.3 | 0.9 | 36.7 | 0.52 | 0.57 | 0.09 |
| !8 Priority Units | 2012–2013 | 0.68 | 0.69 | 4.1 | 5.1 | 3.6 | 3.8 | 3.2 | 2.8 | 78.1 | 0.54 | 0.76 | 0.36 |
| Total | 2012–2013 | 0.43 | 0.43 | 2.2 | 2.7 | 2.0 | 1.9 | 1.7 | 1.4 | 80.4 | 0.35 | 0.47 | 0.18 |

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
