# Peer review of "A Potential Approach of Reporting Risk to Baseflow from Increased Groundwater Extraction in the Murray-Darling Basin, South-Eastern Australia"

_water, doi:10.3390/w14132118_

Round 1

Reviewer 1 Report

Review of “A potential approach to reporting risk to baseflow from increased groundwater extraction in the Murray-Darling Basin south-eastern Australia”

Overview

The paper develops an approach for reporting the impact of increasing groundwater pumping on surface water flows with an aim to identifying regions where groundwater use may threaten flows important for the provision of ecosystem services. 

Major Comments

1.       More focus in intro – what does the paper actually do?  Results from lines 188-194 don’t really align with the approach detailed earlier (lines 166-188) or it’s not immediately apparent to me how they’re related.

2.       Is 4 “Assessing impacts on streamflow” main point of the paper?  It’s not totally clear to me how 1)-5) are connected. “As the aim for this paper is to provide information for the risk analysis” I suggest being very clear in the first few paragraphs precisely the purpose of the paper and then sticking to that message throughout.  As is, the paper reads like a catalogue of the 5 basic building blocks and the reader is left wondering how all 5 are connected. 

3.       Framing/external validity:  while reading this paper I kept wondering why someone not interested in the MDB would be interested in or read the paper?  There is room (and need) to apply the insights of this case study to other regions.  How would this approach work in other contexts?  Make the case that a reader that doesn’t care about the MDB but has an interest in groundwater management should read the paper.

4.       Equations 4 and 5:  I don’t think adding/subtracting standard errors is the best way to go about dealing with the uncertainty about CF values (notation issue: it is not totally clear from your notation with SE refers to – I am guessing SE of CF as that is estimated while E_i and E_f are observed).  Instead, I suggest using 95% confidence intervals to build a distribution of CF and then calculate R_x by taking random draws from the CF distribution.  This will allow you to report confidence intervals for the R_x calculations.

5.       Risk assignment:  Determining the risk of loss of flows seems to be missing information on individuals/societies risk preferences i.e. a risk averse society or policy maker would likely classify more basins as in risk of diminished surface water availability.  Worth discussing. 

6.       A discussion of the importance of groundwater data seems to be missing.  Accurate well-level measurements of water use are vital to the sort of reporting/management regime developed here.  Mentioning this importance is an oversight.  

7.       Line 760: “The analyses showed that about twenty years of annual data is required to develop trends with confidence.”  I don’t see how the analysis supports this claim – proving this would need additional analysis looking at the marginal impact of an additional year of data at increasing the likelihood of identifying a trend.

Minor Comments

1.       I prefer acronyms and symbols I parentheses when first introduced.

2.       I suggest introducing the primary research question (lines 166-188) in first few paragraphs.

3.       Define adaptive management, this an important part of the paper but no definition is provided.

4.       What’s a groundwater unit?  A definition would be helpful.

5.       The intro list 7 “basic building blocks” (page 4, lines 166-188) but the 6th and 7th ‘blocks’ are never mentioned again.  Why?

6.       Some background on property rights of water/legal institutions would be useful in the background section.

7.       Table 1: What are the units of the data?  Caption says volume but it’s unclear what unit.  Also, what does the “remainder” row refer to?

8.       It would be helpful to tell the reader what the LINEST function does.  I am guessing it’s some sort of OLS based linear regression function. 

9.       Figure 2: I suggest removing the lines connecting the dots.  These lines add little to the figure and make it more difficult to interpret.   Could potentially replace with a line of best fit.

10.   Lines 479-515: Back to back 4s.  I think it should be “5. Assignment of Risk”

11.   Figure 3: Why not just label the x-axis with year rather than years since ….

12.   Figure 5b: Change the x axis bounds to only focus on the years included in the regression.

13.   Figure 5b: Differing data shown are not clear e.g. what does multiple regression line represent?  Are these in sample predictions of normalized annual extraction?  If not, then I don’t see why looking at +- SEs makes sense.  Also, the 95% CI would be a more appropriate measure to see accuracy of predictions.

14.   Why 18 priority groundwater units?  Could easily be 10 or 15 or 20, no?  This seems to merit some discussion.

Author Response

The reviewer's comments have been very constructive and useful for the revision. Detailed comments are attached. All but two of the comments have been agreed to. In particular, the structure of the paper has been made more coherent and the statistics of the trend analysis tightened up. This has meant re-doing many of the calculations and graphs. The two comments that had not been accepted are 1) doing a Monte Carlo analysis of the impact 2) redoing the x-axis of the graphs as years rather than years since 2003-4 for reasons outlined.

Reviewer 2 Report

Some items need to be improved are listed as follows:

  1. The figures 2-7 shouled be improved to be more regular.
  2. a methodology chart is suggested to be added. 

Author Response

The comments have been accepted. There has been a major change to the paper and graphs. Detailed comments are attached.

Reviewer 3 Report

The manuscript presents a study of a regional nature (a river basin with extension of the order of a million km2) in an area with a predominantly semi-arid climate where groundwater resources are historically subjected to significant pressure. The study proposes indicators to characterize the risk situation based on groundwater extraction and precipitation records and the analysis of a considerable database and previous hydrological reports. It is a large work due to the area covered and the detail in the description and the good illustrations (tables and figures) together with a long list of references, all very pertinent. The manuscript is easy to read and assimilate, despite the use of many acronyms and its aforementioned length (about 25 pages). It is a remarkable merit of its sole author.

I think it can be useful in areas with similar problems and I recommend its publication in its current state.

Author Response

I would thanks the reviewer for his comments. There does not appear to be any action required. Please note there have been significant changes o the manuscript.